# Where Do Vapers Buy Their Vaping Supplies? Findings from the International Tobacco Control (ITC) 4 Country Smoking and Vaping Survey

**DOI:** 10.3390/ijerph16030338

**Published:** 2019-01-26

**Authors:** David C. Braak, K. Michael Cummings, Georges J. Nahhas, Bryan W. Heckman, Ron Borland, Geoffrey T. Fong, David Hammond, Christian Boudreau, Ann McNeill, David T. Levy, Ce Shang

**Affiliations:** 1Colleges of Graduate Studies and Medicine, Medical University of South Carolina, Charleston, SC 29425, USA; braak@musc.edu; 2Department of Psychiatry & Behavioral Sciences, Medical University of South Carolina, Charleston, SC 29425, USA; elnahas@musc.edu (G.J.N.); heckmanb@musc.edu (B.W.H.); 3Hollings Cancer Center, Medical University of South Carolina, Charleston, SC 29425, USA; 4Nigel Gray Fellowship Group, Cancer Council Victoria, Melbourne 3004, Australia; Ron.Borland@cancervic.org.au; 5School of Psychology, Deakin University, Burwood, Melbourne 3220, Australia; 6Department of Psychology and School of Public Health and Health Systems, University of Waterloo, Waterloo, ON N2L 3G1, Canada; gfong@uwaterloo.ca; 7Department of Psychology, University of Waterloo, Waterloo, ON N2L 3G1, Canada; 8Ontario Institute for Cancer Research, Toronto, ON M5G 0A3, Canada; 9School of Public Health and Health Systems, University of Waterloo, Waterloo, ON N2L 3G1, Canada; david.hammond@uwaterloo.ca; 10Department of Statistics and Actuarial Science, University of Waterloo, Waterloo, ON N2L 3G1, Canada; cboudreau@uwaterloo.ca; 11Department of Addictions, Institute of Psychiatry, Psychology, and Neuroscience, King’s College London, London WC2R 2LS, UK; ann.mcneill@kcl.ac.uk; 12Department of Oncology, Lombardi Comprehensive Cancer Center, Georgetown University, Washington, DC 20057, USA; dl777@georgetown.edu; 13Oklahoma Tobacco Research Center, Stephenson Cancer Center, University of Oklahoma Health Sciences Center, Oklahoma City, OK 73104, USA; ce-shang@ouhsc.edu

**Keywords:** electronic cigarettes, vaping, vaping policies

## Abstract

*Aim:* This study examines where vapers purchase their vaping refills in countries having different regulations over such devices, Canada (CA), the United States (US), England (EN), and Australia (AU). *Methods:* Data were available from 1899 current adult daily and weekly vapers who participated in the 2016 (Wave 1) International Tobacco Control Four Country Smoking and Vaping. The outcome was purchase location of vaping supplies (online, vape shop, other). Adjusted odds ratios and 95% confidence intervals were reported for between country comparisons. *Results:* Overall, 41.4% of current vapers bought their vaping products from vape shops, 27.5% bought them online, and 31.1% from other retail locations. The vast majority of vapers (91.1%) reported using nicotine-containing e-liquids. In AU, vapers were more likely to buy online vs other locations compared to CA (OR = 6.4, 2.3–17.9), the US (OR = 4.1, 1.54–10.7), and EN (OR = 7.9, 2.9–21.8). In the US, they were more likely to buy from vape shops (OR = 3.3, 1.8–6.2) or online (OR = 1.9, 1.0–3.8) vs other retail locations when compared to those in EN. In CA, vapers were more likely to purchase at vape shops than at other retail locations when compared to vapers in EN (5.9, 3.2–10.9) and the US (1.87, 1.0–3.1). *Conclusions:* The regulatory environment and enforcement of such regulations appear to influence the location where vapers buy their vaping products. In AU, banning the retail sale of nicotine vaping products has led vapers to rely mainly on online purchasing sources, whereas the lack of enforcement of the same regulation in CA has allowed specialty vape shops to flourish.

## 1. Introduction

Sales of nicotine vaping products (NVPs), also referred to as electronic cigarettes, have increased over the past decade, and in some countries they have become the most popular quitting aid used by smokers [1,2,3,4]. As vaping prevalence has increased over time, so has the diversity of products available to consumers [5]. NVPs broadly include three types of products: (1) closed systems that are intended to be thrown out when emptied (closed disposable system), (2) rechargeable devices which use pre-filled e-liquid cartridges (closed cartridge system); and (3) and rechargeable open system devices that allow users to refill the e-liquid used (open tank system). Different NVPs differ in how they deliver nicotine, which influences quitting cigarette-smoking [6,7,8,9], though individual factors may contribute as well [10]. Some types of products (ex. tank systems) require more accessory parts than others (ex. disposable), it is possible that some retail locations will be more or less likely to sell certain products. Users of various products may, therefore, be more or less likely to purchase at certain locations as well.

A 2014 survey of online sites selling NVPs identified at least 466 brands available in 7764 assorted flavors, with two types of propellants, offering an average of four to five nicotine strengths [11]. Recent evidence reviews suggest that NVPs are likely to have a substantially lower health impact compared to cigarettes, although these evidence reviews also suggest that nicotine vaping is not done without some potential health risks and the health risks are likely to be different for smokers who vape exclusively compared to those who smoke and vape concurrently [12,13]. Because of their novelty and the lack of conclusive evidence on their health effects, safety, and cessation efficacy, it has been unclear whether NVPs should be regulated as tobacco products, therapeutic goods, medical devices, or consumer lifestyle products [12,13].

Countries have taken different approaches to regulating NVPs, with some banning their sale at retail without pre-market government authorization, and others allowing their sales with specific restrictions, primarily age restrictions. Regulation in Australia (AU), and until recently in Canada, (CA) prohibited the retail sale of NVPs unless government authorization had been obtained prior to marketing. Non-nicotine vaping products were allowed for sale in all four countries. In CA, the law restricting the sale of NVPs in retail locations changed in May 2018. However, at the time of this study in 2016, health authorities in Canada did not enforce the NVP sales ban in retail establishments [14]. NVPs could be legally sold to adults at retail locations in England (EN) and the United States (US). Studies found that government regulations restricting the sale and marketing of NVPs can influence consumer awareness and use of NVPs [4].

While countries regulated the source of purchase differently, information on where individuals buy their NVPs is limited. One cross sectional study on California high school students, found that the most likely purchase location for NVPs was “smoke shops” for those older and younger than 18. This study did not include “vape shop” in its answer choices, was mainly concerned with tobacco products in general, and the majority of students surveyed obtained their products from a friend [15].

In this study, we reported on where current (daily and/or weekly) vapers purchase their vaping products and refills, behavioral characteristics of users, and product features by purchase location. We also compare purchase location by country of residence.

## 2. Methods

### 2.1. Study Participants

Data were available from a sub-sample of the 2016 (Wave 1) International Tobacco Control Four Country Smoking and Vaping (ITC 4CV1) participants who reported currently vaping daily or weekly. Details about this survey can be found elsewhere [16]. Briefly, the ITC four country survey is an online panel-based survey which included respondents 18 years or older who had smoked at least 100 cigarettes in their lifetime, who was currently smoking, or who had quit smoking within the last two years. Recruitment was conducted entirely from web panels. Analyses presented in this paper were restricted to those who reported purchasing their own vaping device or e-liquid. Those who did not remember any details of their last purchase and those who did not answer the purchasing question were excluded. There was a total of 1899 current daily or weekly vapers who reported purchasing their products within the last 30 days, 456 in CA, 701 in the US, 667 in EN, and 75 in AU. Analyses presented in this paper were restricted to those who reported purchasing their own vaping device or e-liquid. Those who did not remember any details of their last purchase and those who did not answer the purchasing question were excluded. There was a total of 1899 current daily or weekly vapers who reported purchasing their products within the last 30 days, 456 in CA, 701 in the US, 667 in EN, and 75 in AU. 

### 2.2. Measures

The outcome measure was where vapers reported purchasing their vaping products (online, vape shop, other location). Participants were asked “Now thinking about the LAST TIME you purchased disposable e-cigarettes, cartridges, or e-liquid, where did you make this last purchase?” Responses were grouped into online, vape shop, or other (i.e., “tobacco specialty shop/tobacconist”, “newsagent/off-license/corner shop/convenience shop”, “convenience store including petrol station”, “gas/petrol station”, “supermarket”, “pharmacy”, “chemist”, “pub or bar”, “temporary mobile sales location”, “some other kind of Shop”, “from outside the country”, or “from somewhere else”). The independent variable was country of purchase (CA, the US, EN, and AU). 

User characteristics included, smoking frequency (i.e., daily, non-daily, not at all), vaping frequency (i.e., daily or weekly), vaping duration (i.e., <1 month, 1–3 months, >3 months). We also examined how demographic characteristics, and vaping product features varied by purchase location in each of the four countries.

Participant demographic characteristics included age (i.e., 18–24 years, 25–39 years, 40–54 years, 55+ years), gender (i.e., male vs female), race (i.e., white or non-white), and educational level (i.e., low = high school or less, moderate = technical degree or some university, high = completed university). In addition, participants in each country were asked, “which of the following categories best describes your annual household income, that is the total income before taxes, or gross income, of all persons in your household combined, for one year?” Because currencies and standards for defining income status varied between countries we categorized respondents as either low, moderate, or high income as follows: US: low = less than 30,000 USD, moderate = 30,000–59,999 USD, and high = 60,000 USD, and AU: low = less than 30,000 AUD, moderate = 30,000–59,999 AUD, and high = 60,000 CAD. For EN, low = less than 30,000 GBP, moderate = 30,000–45,999 GBP, and high = 45,000 GBP or more.

Vaping product characteristics included self-reported e-liquid nicotine strength which was collapsed into three groups: (1) no nicotine, (2) contains nicotine; and (3) don’t know, e-liquid flavor (i.e., tobacco, menthol or mint, fruit, candy/deserts/sweets, and other), product type (i.e., disposable, cartridge, and tank systems), and product modifiability (i.e., power not modifiable, power adjustment possible—but not used, power adjustment possible—and used, and don’t know), following the approach suggested by O’Connor et al [5].

### 2.3. Data Analysis

Weighted generalized logistic regression for survey data (i.e., proc surveylogistic) was performed using SAS 9.4 (SAS Institute, Cary, NC, USA), accounting for missing data being missing not at random (i.e., nomcar option) with purchasing location as the dependent variable and country as the independent variable. The model was adjusted for smoking and vaping frequency, age, gender, ethnicity, household income, educational level, e-liquid flavor, and type of vaping device. Those who refused or did not report ethnicity, household income, and educational level were excluded along with those who reported buying nicotine-free e-liquid. Frequencies and percentages were reported as well as odds ratios with 95% confidence intervals for pair-wise comparisons between countries (i.e. country vs country by location vs location).

### 2.4. Ethics Approval

The survey protocols and all materials, including the survey questionnaires, were cleared for ethics by Institutional Review Board, Medical University of South Carolina (ORE #: 20803); Research Ethics Office, King’s College London, UK (ORE #: 20803 and RESCM-17/18-2240); Office of Research Ethics, University of Waterloo, Canada (ORE #: 20803 and ORE #: 21609); and Human Research Ethics, Cancer Council Victoria, Australia (ORE #: 21609 and HREC 1603).

## 3. Results

Overall, in all four countries the majority of current vapers were white, and male. The ages of the participants were fairly evenly distributed, though in the US and AU the largest age category was 25–39 (US: 33.4%, AU: 40.8%). In England and in Canada the ages of the participants skewed a little older with the largest category in England being 55+ (38.3%) and the largest category in Canada being 40–45 (38.6%). In all four countries, most current vapers were categorized as daily vapers; in EN and AU greater than 50% of current vapers did not smoke at all, whereas in Canada the largest category of smoking frequency was daily smoking (45.2% of current vapers). Current vapers in all four countries reported having a high household income the most often, and a moderate education level (except in the US where the largest category of education level was “low”). In total 41.4% of current vapers bought their vaping products from vape shops, 27.5% bought them online, and 31.1% from other retail locations. The vast majority of vapers (91.1%) reported using nicotine-containing e-liquids; 90.3% in CA, 90.2% in the US, 94.6% in EN, and 88.5% in.

In CA and the US, the majority of vapers bought their vaping products from vape shops, 65.3% and 43.3%, respectively. In EN, most vapers bought their products from other retail locations (37.7%), and in AU, mostly from online sources (65.2%). In all four countries, the majority of NVP users who were purchasing their own products reported using tank system NVPs (67.6% in CA, 59.4% in US, 75.1% in EN, and 82.4% in AU), followed by cartridge systems (19.1% in CA, 29.6% in US, 20.7% in EN, and 10.1% in AU), and disposables (13.2% in CA, 11% in US, 4.2% in EN, and 7.6% in AU). With the exception of NVP users in EN (42.5%), the majority of survey participants reported using NVP that allowed for adjustable power (61.7% in CA, 59.1% in the US, and 57.9% in AU), though this question did not assess the 259 participants who reported using disposable systems. 

Table 1a–d display the bi-variate descriptive characteristics of NVP users by purchase location separately within each of the four countries. Overall, user characteristics and product characteristics were fairly similar across the four countries. 

Table 2 compares the likelihood of purchasing locations of NVPs between all countries. In AU, vapers were 6.4 times more likely to buy online vs other locations compared to vapers in CA (95% CI: 2.3–17.1), 4.1 times more likely than in the US (95% CI: 1.5–10.7), and 7.9 times more likely than respondents in EN (95% CI: 2.9–21.8). In the US, they were 3.3 times more likely to buy from vape shops (95% CI: 1.8–6.2) and 1.9 times more likely to buy online (95% CI: 1.0–3.7) vs other retail locations when compared to those in EN. Additionally, in CA, vapers were 5.1 times more likely to purchase at vape shops than at other retail locations when compared to vapers in EN (95%CI: 3.2–10.9) and 1.8 times more likely than in the US (95% CI: 1.0–3.1).

## 4. Discussion

Differences in how governments regulate and enforce laws regulating the legal points-of-sale of NVPs appeared to influence where vapers obtained their vaping products and supplies. In CA, Vape shops were the most preferred location for products purchase. In the US and EN the purchase locations were fairly evenly distributed among vape shops, online, and other. In AU, most participants reported purchasing their products online. 

In this survey the vaping products that were assessed, aside from the vaping device themselves were refills (i.e. pods, cartridges, e-liquid). In AU, the ban of sale of NVPs appears to have led to current vapers relying mainly on online purchasing sources, unless they are buying non-nicotine containing e-liquid, in which case, other retail outlets were the primary purchase location. The frequent purchase from vape shops in CA was a surprise since technically the sale of NVPs was prohibited in retail outlets without prior government approval. Up to 2016, no NVPs had received approval. It was widely reported that Canadian authorities were not aggressively enforcing the rules preventing the sale of NVPs in retail establishments, which may explain the reporting of vape shops as the primary source of NVPs [14]. The law prohibiting the sale of NVPs in CA appears to have discouraged the marketing of NVPs manufactured by cigarette companies and other vaping manufacturers who primarily market their products in traditional cigarette selling outlets [17]. However, the lax enforcement of the law may have allowed independently owned vape shops to flourish. As of May 2018, NVPs can be legally sold in retail outlets in Canada that is likely to change where Canadian vapers will report buying their NVPs in the future [18]. NVP users in the US and EN reported a wider variety of purchasing sources for their NVPs, although vape shops were a popular location especially for exclusive NVP users using tank systems. 

While it appears that the regulatory environment in different countries can influence vaping prevalence, even in restricted environments, such as AU, vaping is still happening with consumers relying on online sources to purchase their nicotine vaping devices and supplies [4]. What is less clear is how the diversity of NVP points of sale influences vaping and/or smoking behaviors. One might expect that having a greater diversity of purchasing sources would provide more product options for consumers at lower prices, thereby increasing vaping prevalence and perhaps contributing to an accelerated decline in cigarette use. On the other hand, it is possible that by having a greater diversity of NVP selling outlets would have the unfortunate effect of encouraging the uptake of vaping by nonsmokers, including teenagers. The apparent growing popularity of JUUL and other pod type vaping products in the US and more recently in Canada has raised concerns about how NVPs are marketed and to whom they are sold [19,20,21,22,23].

Vape shops permit the purchase of NVPs in an environment where cigarettes are typically not available. Two previous studies have suggested that smokers getting their NVPs from vape shops were more likely to stop smoking cigarettes completely [24,25]. Although, it remains unclear if the devices and information received in vape shops account for the higher quit rates or if it is something about those who purchase vaping products from vape shops which make them more likely to discontinue smoking. It is well established in the smoking cessation treatment literature that combining pharmacotherapy with behavioral counseling increases the odds of smoking cessation over pharmacotherapy or counseling alone [26]. In theory, one might expect that smokers purchasing their NVPs from a vape shop would get better instruction on how to vape to maximize nicotine delivery compared to those purchasing their NVPs online or in a cigarette selling retail store. Additionally, since vape shops are typically not in the business of selling cigarettes or other tobacco products, there would be an incentive for the vape shop owners to encourage consumers to completely switch away from cigarettes [27]. On the other hand, vape shops remain unregulated, so it is unclear if consumers would get accurate information about the NVPs they are purchasing [28]. While vape shops will likely need to be regulated to some extent, the form and extent of regulatory oversight have not yet been defined in the US [29]. 

This study has important limitations to consider. First, this is a descriptive study based on the first wave of data collection making it impossible to draw firm conclusions about the temporality of the associations. Second, this study only reports on the purchasing behaviors of current daily or weekly vapers, whose purchasing patterns may be different for those who vape less frequently or those who did not purchase their products. Third, the vaping product marketplace is rapidly changing so that product features common to those NVP users in 2016 may not hold for those vaping today or in the future. Finally, the small number of participants reporting the use of non-nicotine-containing e-liquid limited the ability to compare them to those who use nicotine-containing liquid, furthermore the “other” location in our analysis was a very heterogeneous group of purchase locations due to low participant counts. 

In summary, the findings from this study show that government regulations restricting where NVPs can be legally sold, and how the regulations are enforced, influence both the location where current NVP users report purchasing their NVPs and the types of product the report purchasing. The AU retail sales ban of NVPs appears to have led vapers to rely on online purchasing sources. In contrast, in CA, the lack of enforcement of this law has allowed vape shops to flourish limiting access to NVPs sold alongside cigarettes. In the US and EN, NVP users reported a greater diversity of purchase locations. Future studies should consider longitudinal data in order to elucidate the effects of policy changes over time. For example, as the US considers limiting the sale of product categories in certain locations, proceeding waves of the ITC project will be well positioned to capture an effect on behavior.

## Figures and Tables

**Table 1 ijerph-16-00338-t001:** Sample characteristics by e-cigarette purchase location in (**a**) Canada (*n* = 456); (**b**) the United States (*n* = 701); (**c**) England (*n* = 667); (**d**) Australia (*n* = 75).

**(a)**				
	**Online**	**Vape Shop**	**Other Retail**	**Total**
	***N* (row %)**	***N* (row %)**	***N* (row %)**	***N* (col %)**
Smoking frequency				
Daily	49 (13.8)	139 (53.6)	87 (32.6)	275 (45.2)
Nondaily	14 (10.6)	79 (67.2)	23 (22.2)	116 (16.9)
Not at all	6 (11.4)	54 (78.5)	5 (10.2)	65 (37.9)
Vaping frequency				
Daily	37 (10.9)	152 (72.8)	51 (16.3)	240 (59.4)
Nondaily	32 (14.4)	120 (54.4)	64 (31.2)	216 (40.6)
Been vaping for				
Less than 1 month	3 (9.2)	16 (75.9)	6 (15)	25 (5.5)
1–3 months	9 (9.3)	39 (65.9)	17 (24.8)	65 (11.5)
>3 months	55 (12.6)	217 (64.8)	92 (22.6)	364 (82.7)
No answer	2 (100)	0 (0)	0 (0)	2 (0.3)
Age				
18–24	18 (14.6)	58 (61.5)	27 (23.9)	103 (15.4)
25–39	29 (19.9)	100 (58.8)	36 (21.3)	165 (30.3)
40–45	19 (9.0)	84 (69.1)	38 (21.9)	141 (38.6)
55+	3 (3.8)	30 (72.3)	14 (23.9)	47 (15.7)
Sex				
Female	29 (11.1)	131 (71.2)	47 (17.7)	207 (46.5)
Male	40 (13.4)	141 (60.2)	68 (26.4)	249 (53.5)
Race				
White	43 (10.1)	227 (67.4)	89 (22.5)	359 (86)
Nonwhite	26 (26.4)	43 (51.7)	26 (21.8)	95 (13.8)
Refused	0 (0)	2 (100)	0 (0)	2 (0.2)
Don’t know	0 (0)	0 (0)	0 (0)	0 (0) (0)
Household Income				
Low	13 (15.6)	53 (61.7)	25 (22.6)	91 (18.7)
Moderate	20 (8.2)	81 (67.5)	34 (24.3)	135 (29.3)
High	36 (15.4)	116 (62.9)	50 (21.7)	202 (45.5)
No answer	0 (0)	22 (82.6)	6 (17.4)	28 (6.5)
Educational level				
Low	10 (4.9)	81 (69.2)	31 (26)	122 (30)
Moderate	27 (10.2)	120 (71)	42 (18.7)	189 (45.2)
High	32 (25.4)	69 (49.7)	42 (24.8)	143 (24.5)
No answer	0 (0)	2 (100)	0 (0)	2 (0.3)
E-liquid flavor				
Tobacco flavor	16 (10.7)	59 (64.2)	33 (25)	108 (22.3)
Menthol or mint	6 (4.9)	38 (53.9)	18 (41.3)	62 (15.2)
Fruit flavor	22 (17.2)	89 (68.3)	27 (14.5)	138 (31)
Candy, desserts, sweets	5 (4.6)	45 (82.2)	9 (13.2)	59 (12.2)
Other	20 (17)	41 (60.2)	28 (22.7)	89 (19.3)
E-liquid containing nicotine				
No	2 (2.6)	14 (71.9)	8 (25.4)	24 (7.8)
Yes	65 (12.4)	254 (65.5)	105 (22.2)	424 (90.3)
Refused	0 (0)	1 (100)	0 (0)	1 (0.1)
Don’t know	2 (52.8)	3 (26.5)	2 (20.7)	7 (1.8)
Product type				
Disposable	9 (13.1)	12 (17.4)	40 (69.4)	61 (13.2)
Cartridge	24 (15.2)	61 (61)	32 (23.8)	117 (19.1)
Tank	36 (11.4)	199 (75.9)	43 (12.7)	278 (67.6)
Product power modifiability				
Missing	9 (13.1)	12 (17.4)	40 (69.4)	61 (13.2)
Power not adjustable	13 (9.9)	62 (72.6)	21 (17.5)	96 (22)
Power adjustable but I don’t change it	23 (16.2)	78 (65.1)	27 (18.8)	128 (26.6)
Power adjustable and I change it	23 (10.9)	110 (77.1)	26 (12)	159 (35.1)
Don’t know	1 (9.7)	10 (86.2)	1 (4.1)	12 (3.1)
Total	69 (12.3)	272 (65.3)	115 (22.4)	456
**(b)**				
	**Online**	**Vape Shop**	**Other Retail**	**Total**
	***N* (row %)**	***N* (row %)**	***N* (row %)**	***N* (col %)**
Smoking frequency				
Daily	107 (23.4)	168 (36.8)	190 (39.7)	465 (40.1)
Nondaily	33 (28.7)	61 (48.9)	38 (22.4)	132 (13.2)
Not at all	22 (29.2)	52 (47.3)	30 (23.5)	104 (46.7)
Vaping frequency				
Daily	111 (27.7)	198 (45.6)	159 (26.8)	468 (78.2)
Nondaily	51 (23.7)	83 (35.3)	99 (40.9)	233 (21.8)
Been vaping for				
Less than 1 month	27 (22.2)	30 (53.4)	43 (24.4)	100 (13.2)
1–3 months	14 (10.4)	32 (35.5)	34 (54)	80 (7.1)
>3 months	120 (29.0)	219 (42.5)	178 (28.5)	517 (79.4)
No answer	1 (44.5)	0 (0)	3 (55.5)	4 (0.3)
Age				
18–24	33 (27.8)	71 (48.2)	65 (24)	169 (16)
25–39	78 (24.4)	120 (43.9)	120 (31.7)	318 (33.4)
40–45	28 (29)	36 (40.8)	30 (30.2)	94 (27)
55+	23 (27)	54 (42.2)	43 (30.8)	120 (23.5)
Sex				
Female	70 (26.3)	124 (45.8)	94 (27.9)	288 (42.4)
Male	92 (27.2)	157 (41.5)	164 (31.3)	413 (57.6)
Race				
White	128 (27.3)	236 (44.5)	203 (28.2)	567 (89.8)
Nonwhite	34 (22.5)	43 (32.8)	55 (44.7)	132 (10.1)
Refused	0 (0)	2 (100)	0 (0)	2 (0.1)
Don’t know	0 (0)	0 (0)	0 (0)	0 (0) (0)
Household Income				
Low	31 (31.3)	58 (43.3)	55 (25.5)	144 (32.8)
Moderate	36 (19)	74 (43.7)	54 (37.3)	164 (23.5)
High	95 (28.5)	145 (42.2)	146 (29.2)	386 (42.3)
No answer	0 ((.))	4 (72.1)	3 (27.9)	7 (1.3)
Educational level				
Low	43 (33.8)	72 (41.8)	55 (24.4)	170 (46)
Moderate	43 (24.6)	86 (46.9)	65 (28.6)	194 (26.4)
High	76 (17.3)	123 (42.5)	138 (40.2)	337 (27.6)
No answer	0 (0)	0 (0)	0 (0)	0 (0) (0)
E-liquid flavor				
Tobacco flavor	44 (30.4)	48 (28.5)	82 (41.1)	174 (26.1)
Menthol or mint	24 (18.2)	58 (38.6)	57 (43.3)	139 (14)
Fruit flavor	29 (20.9)	76 (55)	46 (24)	151 (29.3)
Candy, desserts, sweets	15 (37.7)	35 (53.5)	10 (8.8)	60 (13.9)
Other	50 (29.7)	64 (41.5)	63 (28.8)	177 (16.7)
E-liquid containing nicotine				
No	5 (47.8)	12 (49.9)	2 (2.3)	19 (3.3)
Yes	156 (27.6)	263 (45.1)	231 (27.3)	650 (90.2)
Refused	0 (0)	0 (0)	1 (100)	1 (0)
Don’t know	1 (5.3)	6 (16.5)	24 (78.2)	31 (6.5)
Product type				
Disposable	35 (34.9)	21 (19)	60 (46.2)	116 (11)
Cartridge	55 (24.6)	77 (16.7)	148 (58.7)	280 (29.6)
Tank	72 (26.4)	183 (61.1)	50 (12.5)	305 (59.4)
Product power modifiability				
Missing	35 (34.9)	21 (19)	60 (46.2)	116 (11)
Power not adjustable	26 (23.8)	45 (29.8)	67 (46.4)	138 (29.1)
Power adjustable but I don’t change it	38 (20.7)	73 (63.4)	46 (15.9)	157 (19.4)
Power adjustable and I change it	61 (29.9)	140 (50.4)	79 (19.7)	280 (39.7)
Don’t know	2 (19.4)	2 (30.6)	6 (50)	10 (0.8)
Total	162 (26.8)	281 (43.3)	258 (29.9)	701
**(c)**				
	**Online**	**Vape Shop**	**Other Retail**	**Total**
	***N* (row %)**	***N* (row %)**	***N* (row %)**	***N* (col %)**
Smoking frequency				
Daily	100 (26.2)	124 (31.3)	130 (42.5)	354 (23.7)
Nondaily	55 (23.3)	61 (40.7)	58 (36)	174 (12.5)
Not at all	52 (36)	40 (27.7)	47 (36.3)	139 (63.8)
Vaping frequency				
Daily	157 (33.6)	160 (29.9)	160 (36.6)	477 (86)
Nondaily	50 (23)	65 (32.1)	75 (45)	190 (14)
Been vaping for				
Less than 1 month	13 (39.2)	11 (37.6)	11 (23.3)	35 (3.3)
1–3 months	23 (43)	30 (17.5)	30 (39.5)	83 (10.7)
>3 months	171 (30.6)	183 (31.2)	194 (38.2)	548 (85.8)
No answer	0 (0)	1 (100)	0 (0)	1 (0.3)
Age				
18–24	61 (22)	80 (56.2)	54 (21.8)	195 (7.1)
25–39	49 (26.1)	61 (32.7)	73 (41.2)	183 (25.6)
40–45	37 (34.5)	37 (26.1)	47 (39.3)	121 (29)
55+	60 (36.2)	47 (26.7)	61 (37.1)	168 (38.3)
Sex				
Female	77 (30.8)	79 (33.7)	80 (35.5)	236 (48)
Male	130 (33.3)	146 (26.9)	155 (39.8)	431 (52)
Race				
White	176 (31)	198 (30.2)	213 (38.9)	587 (92.1)
Nonwhite	24 (49.4)	21 (18.2)	18 (32.3)	63 (5.2)
Refused	1 (17.3)	2 (68)	2 (14.7)	5 (1.2)
Don’t know	6 (54.9)	4 (41.7)	2 (3.4)	12 (1.5)
Household Income				
Low	26 (18.6)	39 (32.3)	62 (49)	127 (17.3)
Moderate	59 (30.7)	62 (32.1)	66 (37.2)	187 (31.7)
High	107 (37)	106 (26.4)	97 (36.7)	310 (42.9)
No answer	15 (40.7)	18 (37.9)	10 (21.3)	43 (8.1)
Educational level				
Low	41 (29.8)	54 (35.3)	48 (34.9)	143 (19.9)
Moderate	83 (29.7)	87 (29.2)	102 (41.1)	272 (60.2)
High	79 (48.2)	81 (23.9)	82 (27.9)	242 (15.7)
No answer	4 (18.1)	3 (42.8)	3 (39.1)	10 (4.3)
E-liquid flavor				
Tobacco flavor	69 (35.2)	56 (23.2)	82 (41.6)	207 (33.8)
Menthol or mint	38 (29.8)	41 (20.5)	53 (49.7)	132 (26.7)
Fruit flavor	58 (28.5)	87 (50.6)	51 (21)	196 (26.1)
Candy, desserts, sweets	9 (16.8)	16 (29.8)	13 (53.4)	38 (3.9)
Other	33 (43.9)	25 (26.3)	36 (29.9)	94 (9.5)
E-liquid containing nicotine				
No	7 (70)	4 (18.5)	3 (11.5)	14 (3.2)
Yes	193 (31.0)	219 (31.0)	217 (38.0)	629 (94.6)
Refused	2 (21.4)	0 (0)	2 (78.6)	4 (0.4)
Don’t know	5 (24.4)	2 (14.2)	13 (61.4)	20 (1.8)
Product type				
Disposable	19 (18.2)	17 (21)	34 (60.9)	70 (4.2)
Cartridge	67 (36.6)	44 (12.2)	83 (51.2)	194 (20.7)
Tank	121 (31.7)	164 (35.6)	118 (32.7)	403 (75.1)
Product power modifiability				
Missing	19 (18.2)	17 (21)	34 (60.9)	70 (4.2)
Power not adjustable	82 (33.5)	66 (24.3)	95 (42.1)	243 (51.7)
Power adjustable but I don’t change it	43 (32.3)	55 (28.9)	41 (38.8)	139 (14.9)
Power adjustable and I change it	57 (31)	84 (43.1)	59 (25.9)	200 (27.6)
Don’t know	6 (40.4)	3 (31.3)	6 (28.3)	15 (1.6)
Total	207 (32.1)	225 (30.2)	235 (37.7)	667
**(d)**				
	**Online**	**Vape Shop**	**Other Retail**	**Total**
	***N* (row %)**	***N* (row %)**	***N* (row %)**	***N* (col %)**
Smoking frequency				
Daily	15 (44.3)	6 (8.5)	21 (47.2)	42 (32.6)
Nondaily	8 (73.8)	0 (0)	4 (26.2)	12 (10.5)
Not at all	15 (75.5)	6 (24.5)	0 (0)	21 (57)
Vaping frequency				
Daily	32 (67.9)	12 (20.1)	16 (12)	60 (83)
Nondaily	6 (52.1)	0 (0)	9 (47.9)	15 (17)
Been vaping for				
Less than 1 month	3 (63.6)	1 (5.3)	4 (31.1)	8 (6.4)
1–3 months	3 (33.3)	1 (7.5)	7 (59.2)	11 (12.6)
>3 months	32 (86.6)	10 (19.0)	14 (10.7)	56 (80.9)
No answer	0 (0)	0 (0)	0 (0)	0 (0)
Age				
18–24	1 (21.1)	2 (71.3)	1 (7.7)	4 (10.6)
25–39	12 (66.1)	3 (14.2)	7 (19.6)	22 (40.8)
40–45	17 (64.3)	5 (9.2)	13 (26.4)	35 (27.5)
55+	8 (86.6)	2 (3.8)	4 (9.6)	14 (21.1)
Sex				
Female	7 (55.8)	8 (30.7)	11 (13.5)	26 (42.2)
Male	31 (72)	4 (6.5)	14 (21.5)	49 (57.8)
Race				
White	35 (70.8)	8 (17.3)	18 (11.9)	61 (88.2)
Nonwhite	3 (22.9)	4 (12.5)	7 (64.6)	14 (11.8)
Refused	0 (0)	0 (0)	0 (0)	0 (0) (0)
Don’t know	0 (0)	0 (0)	0 (0)	0 (0) (0)
Household Income				
Low	5 (80.1)	0 (0)	4 (19.9)	9 (8.1)
Moderate	7 (72.8)	2 (11.2)	6 (16)	15 (28.9)
High	26 (63.4)	9 (21.9)	14 (14.7)	49 (59.5)
No answer	0 (0)	1 (13)	1 (87)	2 (3.6)
Educational level				
Low	11 (66.1)	3 (27)	3 (6.9)	17 (32.6)
Moderate	12 (65.9)	4 (11.5)	10 (22.6)	26 (43.8)
High	15 (62.5)	5 (12.1)	12 (25.3)	32 (23.6)
No answer	0 (0)	0 (0)	0 (0)	0 (0) (0)
E-liquid flavor				
Tobacco flavor	14 (72.8)	1 (2.1)	7 (25.1)	22 (16.3)
Menthol or mint	1 (36.1)	2 (13)	5 (51)	8 (7.2)
Fruit flavor	6 (40.2)	3 (36.1)	5 (23.7)	14 (20)
Candy, desserts, sweets	5 (66.3)	2 (33.7)	0 (0)	7 (18.2)
Other	12 (80)	4 (5.4)	8 (14.6)	24 (38.3)
E-liquid containing nicotine				
No	1 (8.1)	1 (10.6)	6 (81.3)	8 (6.5)
Yes	34 (68.6)	11 (18.1)	17 (13.3)	62 (88.5)
Refused	0 (0)	0 (0)	0 (0)	0 (0) (0)
Don’t know	3 (78.5)	0 (0)	2 (21.5)	5 (5)
Product type				
Disposable	2 (10.3)	0 (0)	10 (89.7)	12 (7.6)
Cartridge	6 (40.2)	4 (14.6)	8 (45.2)	18 (10.1)
Tank	30 (73.2)	8 (18.5)	7 (8.3)	45 (82.4)
Product power modifiability				
Missing	2 (10.3)	0 (0)	10 (89.7)	12 (7.6)
Power not adjustable	11 (75.2)	2 (3.1)	9 (21.8)	22 (33.7)
Power adjustable but I don’t change it	8 (88)	2 (9.6)	1 (2.5)	11 (10.2)
Power adjustable and I change it	16 (61.3)	8 (30.8)	5 (7.9)	29 (47.7)
Don’t know	1 (100)	0 (0)	0 (0)	1 (0.9)
Total	38 (65.2)	12 (16.7)	25 (18.1)	75

**Table 2 ijerph-16-00338-t002:** Adjusted analyses of purchase location (*n* = 1508).

	Online vs Other	Vape Shop vs Other	Vape Shop vs Online
	AOR (95% CI)	AOR (95% CI)	AOR (95% CI)
Australia vs Canada	6.4 (2.3–17.9) *	0.3 (0.1–1.1)	0.1 (0.01–0.2) *
Australia vs England	7.9 (2.9–21.8) *	2.0 (0.6–6.6)	0.2 (0.1–0.9) *
Australia vs United States	4.1 (1.5–10.7) *	0.6 (0.2–2.1)	0.1 (0.1–0.6) *
Canada vs England	1.2 (0.6–2.6)	5.9 (3.2–10.9) *	4.8 (2.4–9.3) *
Canada vs United States	0.6 (0.3–1.3)	1.8 (1.0–3.1)	2.8 (1.5–5.3) *
England vs United States	0.5 (0.3–1.0)	0.3 (0.2–0.6) *	0.6 (0.3–1.1)

Adjusted for smoking status, vaping status, age, sex, ethnicity, education, income, e-liquid flavor, and device type; excludes race (refused, don’t know), educational level (no answer), household income (no answer), and nicotine concentration (0, refused, don’t know). * indicates 95% CI does not include 1.0.

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
