# Peer review of "Where Do Vapers Buy Their Vaping Supplies? Findings from the International Tobacco Control (ITC) 4 Country Smoking and Vaping Survey"

_ijerph, 2019, doi:10.3390/ijerph16030338_

Round 1

Reviewer 1 Report

Dear Authors,

I've read your paper with interest and and I highly score significance of its content.

I have some commentaries to the text:

Introduction:

In my opinion this paragraph lacks statement on described toxicity of vapour coming form e-cigarettes. There is sufficient data to suggest that although e-cigarettes are less toxic that tobacco cigarettes they are not safe.

Methods:

Study participants section lacks brief info on recruitment process. You mention about other paper where readers can find info but readers should be informed in this paper some basic facts on that to assess the validity of this process.

line 105 - clarify, please is this income for person in household/year? be specific

Results

As you present data in table 1a-d as bivariate descriptive characteristic should you add info on at least chi2 analysis, to let readers know if results are significant? If not only table 2 presents significant results - so it is a bit little for such paper - unless you present it only as a brief report.

Discussion

Please update discussion comparing data on access to electronic cigarettes in other countries  

Author Response

Point-by-point response to reviewer comments

Comment:  The authors should add a statement on the health risks of e-cigarettes.

Response:  We have added the following two sentences to the end of paragraph 2:

Recent evidence reviews suggest that NVPs are likely to have a substantially lower health impact compared to cigarettes, although these evidence reviews also suggest that nicotine vaping is not done without some potential health risks and the health risks are likely to be different for smokers who vape exclusively compared to those who smoke and vape concurrently.12, 13 Because of their novelty and the lack of conclusive evidence on their health effects, safety, and cessation efficacy, it has been unclear whether NVPs should be regulated as tobacco products, therapeutic goods, medical devices, or consumer lifestyle products.12, 13

Comment:  In addition to referencing the methods paper, the authors should provide readers with a brief description of how the how respondents were identified and selected for inclusion in the survey.

Response:  In the methods section of the paper we have added the following information on how the study sample of selected:

Briefly, the ITC four country survey is an online panel-based survey which included respondents 18 years or older who had smoked at least 100 cigarettes in their lifetime, who was currently smoking, or who had quit smoking within the last two years. Recruitment was conducted entirely from web panels.

Comment: The authors in the results should clarify how household income was assessed

Response: The following paragraph was altered to include the currency abbreviations and income distribution categories for each country in the analysis. In addition a statement on the method by which income was assessed was added to the end of the paragraph.

In addition, participants in each country were asked, “which of the following categories best describes your annual household income, that is the total income before taxes, or gross income, of all persons in your household combined, for one year?”  Because currencies and standards for defining income status varied between countries we categorized respondents as either low, moderate, or high income as follows: US: low = less than 30000 USD, moderate = 30000-59999 USD, and high = 60000 USD, and AU: low = less than 30000 AUD, moderate = 30000-59999 AUD, and high = 60000 CAD. For EN, low = less than 30000 GBP, moderate = 30000-45999 GBP, and high = 45000 GBP or more.

Comment:  As you present data in table 1a-d as bivariate descriptive characteristic should you add info on at least chi2 analysis, to let readers know if results are significant? If not only table 2 presents significant results - so it is a bit little for such paper - unless you present it only as a brief report.

Response: Tables 1a-d are meant to be descriptive tables of the demographics of the sample in each country. Although chi-square analysis would be able to demonstrate if differences within each country existed, it would not show us where those differences were, because there were at least 3 dimensions in the outcome variable. Further, the focus of this paper was on the effect of policy differences between countries, thus we chose to limit our statistical analysis to between country comparisons. We hesitate to include numerous chi-square tests, because they would not be any more specifically descriptive than the point estimates, the numerous tests would add uncertainty to our paper, and they would not improve our conclusions on the effects of government regulation on consumer purchasing decisions.

Comment: Update the discussion comparing data on access to electronic cigarettes in other countries.

Response: We are not exactly clear what the reviewers are requesting.  We are assuming the reviewer was referencing the May 2018 change in the regulations governing the sale of NVPs in Canada.  Below is what we say about the regulations governing the sale of NVP in retail locations in the four countries:

As of May 2018, NVPs can be legally sold in retail outlets in Canada that is likely to change where Canadian vapers will report buying their NVPs in the future.

Reviewer 2 Report

The study presented is the second part of another published study. The manuscript is understandable, it contains all the elements of a cross-sectional study. The introduction is complete as well as the results and discussion, mentioning the limitations. However, although it is mentioned that the details of the survey are published in another study (reference 16), it would be important to briefly explain some details, for example if the survey was by telephone, or by mail, the selection of the sample , etc. All this would improve comprehension and reading.

Author Response

Thank you for your comments. In the methods section of the paper we have added the following information on how the study sample of selected:

Briefly, the ITC four country survey is an online panel-based survey which included respondents 18 years or older who had smoked at least 100 cigarettes in their lifetime, who was currently smoking, or who had quit smoking within the last two years. Recruitment was conducted entirely from web panels.